# Practical uncertainty quantification for brain tumor segmentation

**Moritz Fuchs**[1]                                    MORITZ.FUCHS@GRIS.INFORMATIK.TU-DARMSTADT.DE
**Camila González**[1]                              CAMILA.GONZALEZ@GRIS.INFORMATIK.TU-DARMSTADT.DE
**Anirban Mukhopadhyay**[1]   ANIRBAN.MUKHOPADHYAY@GRIS.INFORMATIK.TU-DARMSTADT.DE
[1] *Technical University of Darmstadt, Karolinenpl. 5, 64289 Darmstadt, Germany*

**Editors:** Under Review for MIDL 2022

## Abstract

Despite U-Nets being the de-facto standard for medical image segmentation, researchers have identified shortcomings of U-Nets, such as overconfidence and poor out-of-distribution generalization. Several methods for uncertainty quantification try to solve such problems by relying on well-known approximations such as Monte-Carlo Drop-Out, Probabilistic U-Net, and Stochastic Segmentation Networks. We introduce a novel multi-headed Variational U-Net. The proposed approach combines the global exploration capabilities of deep ensembles with the out-of-distribution robustness of Variational Inference. An efficient training strategy and an expressive yet general design ensure superior uncertainty quantification within a reasonable compute requirement. We thoroughly analyze the performance and properties of our approach on the publicly available BRATS2018 dataset. Further, we test our model on four commonly observed distribution shifts. The proposed approach has good uncertainty calibration and is robust to out-of-distribution shifts.

**Keywords:** uncertainty estimation, out-of-distribution, brain tumor segmentation

## 1. Introduction

While standard U-Nets have become the backbone architecture for most medical image segmentation problems (Bakas et al., 2018; Ronneberger et al., 2015), their uncertainty-aware interpretations are gaining attention within the medical imaging community (Kohl et al., 2018; Kwon et al., 2020; Senapati et al., 2020; Gonzalez et al., 2021). As most state-of-the-art segmentation models in the medical field are overconfident and fail to segment slightly out-of-distribution images, an uncertainty-aware model provides an intuitive way to quantify the inherent uncertainty of the segmentation. Critical downstream tasks such as surgical or radiation therapy planning can benefit from quantifying segmentation uncertainty by becoming more robust against the possibility of damaging critical risk structures. In addition, giving the clinician direct feedback about the uncertainty of a prediction will increase the reliability and raise the trustworthiness of an AI solution. For sufficient feedback to a clinician, we need to quantify the epistemic uncertainty as this provides direct feedback about the systematic uncertainty the AI solution has given a sample. However, high-quality pixel-level annotation is prohibitively expensive in the medical setting (Yushkevich et al., 2019). As a result, annotations are often done by a single expert. Therefore, U-Nets trained on a single annotator typically overfit and miss-segment slightly out-of-distribution (OOD) images (e.g. due to acquisition issues or motion artifacts) that are relatively common in

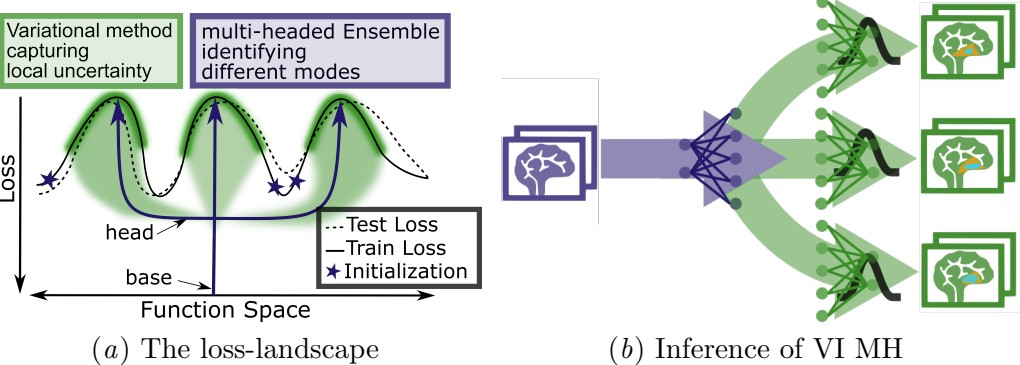

(a) The loss-landscape          (b) Inference of VI MH

Figure 1: (a) The loss-landscape shows how the VI MH converges to separate local optima. (b) Flow-diagram for Inference with VI MH. The base model extracts image features, while the heads are variational functions predicting different modes.

medical imaging (Dou et al., 2019). Reliable uncertainty estimation is fundamental to increase the robustness and trust of the model (Guo et al., 2017; Kendall and Gal, 2017).

We propose a specialized multi-headed Variational U-Net architecture (see Figure 1) that combines the global exploration power of Deep Ensembles with the out-of-distribution robustness of Variational Inference (VI). While multi-headed U-Net (MH) approximates the benefits of deep ensembling, the design of shared weights in the base model ensures efficient computation. The proposed architecture is both compute and space-efficient, yet expressive enough to approximate the uncertainty of the prediction. However, a simple multi-headed U-Net can only perform a Maximum-a-Posteriori estimation at each ensemble head. We circumvent this problem by approximating each local optima with a parameterized posterior see Figure 1(a)). For this purpose, we analyze different models and propose a specialized multi-headed Variational (VI MH) U-Net architecture (see Figure 1(b)) that is an efficient yet expressive uncertainty-aware interpretation of U-Net. We verify our approach on the BRATS2018 (Bakas et al., 2017, 2018; Menze et al., 2014) dataset and evaluate OOD performance on four distribution shifts generated with the TorchIO library (Pérez-García et al., 2021).

## 2. Related Work

The overwhelming majority of uncertainty-aware neural networks rely on the Monte-Carlo Drop-Out (MCDO) approximation (Gal and Ghahramani, 2016; Kingma et al., 2015; Zhang et al., 2018). While using the Drop-Out function is inexpensive in terms of computation, there are several inherent limitations of MCDO, including less reliable uncertainty quantification, poor calibration, and bad out-of-distribution detection (Caldeira and Nord, 2020; Gal et al., 2017; Ovadia et al., 2019; Folgoc et al., 2021). Furthermore, a high Drop-Out rate in MCDO yields worse performance and expected calibration error (Ovadia et al., 2019).

Another popular method is mean-field Variational Inference (VI) (Blundell et al., 2015; Graves, 2011; Kingma et al., 2015). VI learns a Gaussian distribution over the weights for each parameter instead of a point estimate. VI shows promising results on out-of-distribution and epistemic uncertainty quantification (Caldeira and Nord, 2020; Ovadia

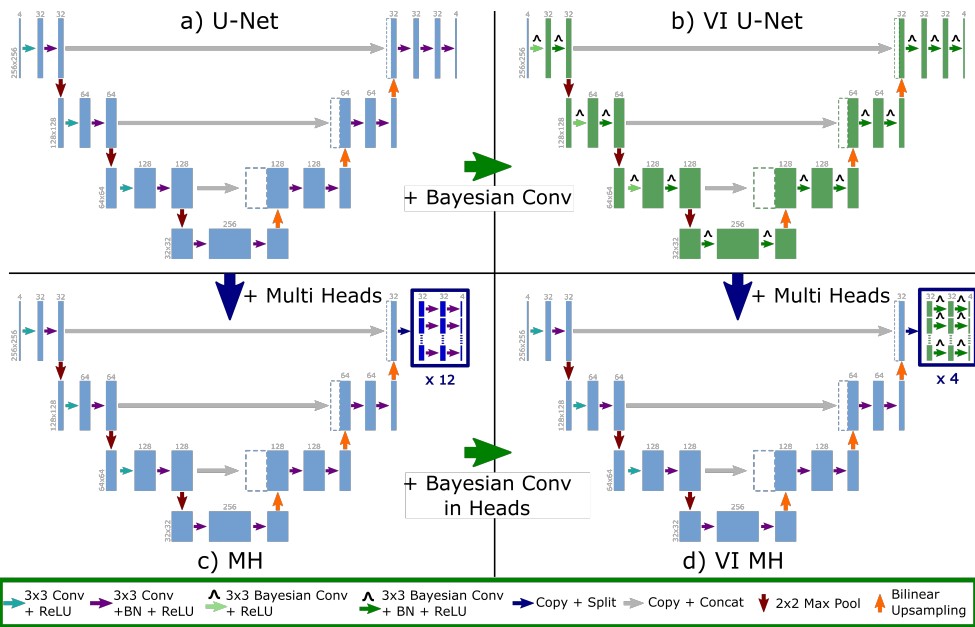

Figure 2: Employing Bayesian convolutions (b) and multi-heads (c) to convert a simple U-Net (a) into VI MH (d). (zoom for details)

et al., 2019). However, VI is often unstable in training and lacks prediction performance (Ovadia et al., 2019; Wenzel et al., 2020).

Meanwhile, Deep Ensembles (Lakshminarayanan et al., 2017) have excellent prediction performance for uncertainty-aware deep learning, provide a reliable out-of-distribution quantification, and are capable of finding different modes within the function space (Fort et al., 2019). Recent work have proposed more efficient implementations like EnsembleNet (Li et al., 2019) and Batch-ensemble (Wen et al., 2019). However, those parameter-sharing approaches have not been shown to represent a divers set of functions within the function space.

The most prominent approaches for uncertainty estimation in segmentation beyond MCDO, VI, and Ensembles are Probabilistic U-Net (Prob. U-Net) (Kohl et al., 2018), PhiSeg (Baumgartner et al., 2019) and Stochastic Segmentation Networks (SSNs) (Monteiro et al., 2020). While SSNs use a low-rank approximation of the covariance matrix for all labels, Prob. U-Net and PhiSeg model a prior within the network architecture. SSNs do generate diverse samples but can only estimate the aleatoric uncertainty. However, Prob. U-Net, PhiSeg and SSNs training procedures are precarious and produce samples with limited diversity when trained with annotation from a single grader. This paper is mainly interested in modeling epistemic uncertainty beyond what the Probabilistic U-Net and PhiSeg are able to capture. The advantage of our VI MH is the better segmentation performance than other methods in a single grader environment, while the VI helps significantly with the uncertainty prediction.

## 3. Method

In this work, we deploy the efficient global exploration of EnsembleNet (Li et al., 2019) on a U-Net-based segmentation architecture for the BRATS2018 dataset. By applying a VI approach on the heads, we improve the model calibration and uncertainty estimation. To better understand these components of our model, we take a look at each method individually.

### 3.1. Variational Inference

The objective of VI is to calculate the posterior distribution of the weights with respect to the data $P(w|\mathcal{D})$ parameterized by a simple distribution $q(\theta)$. We assume this distribution to be Gaussian $\mathcal{N}(0, 0.1 * \mathbf{I})$. For this kind of Bayesian neural network (BNN) the prediction $\hat{y}$ is given by the expectation $\mathbb{E}_{P(w|\mathcal{D})}[P(\hat{y}|\hat{x}, w)]$. This yields the prediction for each test sample $\hat{x}$ for all possible weight configurations in the posterior distribution. Rather than sampling the posterior distribution, it is more efficient to use the local re-parameterization trick introduced by Kingma et al. (2015) since it helps factor out the noise in the back-propagation: $w = \mu_\theta + \sigma_\theta \epsilon$ and $\epsilon \sim \mathcal{N}(0, \mathbf{I})$.

The local re-parameterization allows us to efficiently compute a less noisy gradient sampling from $\mathcal{N}(\mu_\theta, \sigma_\theta)$. The $\sigma_\theta$ needs to be represented as a positive value. Therefore we use the softplus function encoding the parameter $\sigma_\theta$ in the weight $\rho$: $\sigma_\theta = \log(1 + e^\rho)$. This trick was proposed by Blundell et al. (2015). To train a BNN, we want to minimize the Kullback-Leibler (KL) divergence with the posterior on the weights. This can be written as the evidence lower bound: $\theta^* = arg\min_\theta KL[q(w|\theta)||P(w|\mathcal{D})]$

$= arg\min_\theta T * KL[q(w|\theta)||P(w)] - \mathbb{E}_{q(w|\theta)}[\log P(\mathcal{D}|w)]$. The temperature is $T = 1$ for the Bayesian posterior. However, empirical works, like (Wenzel et al., 2020; Zhang et al., 2018), suggest to set this temperature at $T << 1$ for best performance.

### 3.2. The multi-headed Ensemble

In contrast to VI, Deep Ensembles sample not on the posterior distribution over the weights but rather on the parameters of the entire posterior. Fort et al. (2019) show that even though different ensemble members converge to different local optima in the loss-landscape, they do perform very similarly in terms of accuracy (See Appendix B for MH). This property lets us view Deep Ensembles as a technique for Bayesian model averaging, where each ensemble member does a Maximum-a-Posteriori estimation. However, Deep Ensembles have the drawback that multiple models must be trained and stored independently. This property linearly increases training time and limits inference speed.

We use a multi-headed model consisting of a base and multiple head models to accommodate this. The depth of a head is a tradeoff with the number of heads $H$ due to memory restrictions. The result is depicted in Figure 2c. To improve the model's capability to capture epistemic uncertainty, we propose the VI MH as illustrated in Figure 2d. We depict all architectures in Figure 2. The base model is a reduced parameter version of the U-Net (Ronneberger et al., 2015). For the VI U-Net model, we replace the regular convolutions with Bayesian convolutions. The same distribution over the weights assumed in the fully connected case for Bayesian convolutions. As opposed to that, VI MH only uses the

Bayesian convolution in the head models. This lets us calculate two losses for the VI MH. The first loss is on the parameters $\Theta := \{\theta_1, ..., \theta_H\}$ of all ensemble heads:

$$\mathcal{L}(\mathcal{D}|\Theta) = T * KL[q(w|\Theta)||P(w)] - \mathbb{E}_{q(w|\Theta)}[\log P(\mathcal{D}|w)] \tag{1}$$

and the second loss for each ensemble head:

$$\mathcal{L}_h(\mathcal{D}|\theta_h) = T * KL[q(w|\theta_h)||P(w)] - \mathbb{E}_{q(w|\theta_h)}[\log P(\mathcal{D}|w)] \tag{2}$$

where $\theta_h$ denotes the parameters of each head $h \in \{1, ..., H\}$. For the MH, we do not assume a prior distribution over the weights, and therefore the KL term is zero. The weighting between the losses is determined by $\lambda$.

$$\mathcal{L}_c := \lambda\mathcal{L}(\mathcal{D}|\Theta) + (1 - \lambda) \sum_{h=1}^{H} \frac{1}{H} \mathcal{L}_h(\mathcal{D}|\theta_h) \tag{3}$$

We interpret $\lambda$ as a controlling parameter between classical Deep Ensemble training and the combined prediction of the network. VI introduces twice as many parameters as a standard U-Net. This results in the forward pass taking approximately twice the run time. However, our MH architecture reduces the number of parameters and speeds up training compared to the traditional ensemble method.

## 4. Results

Table 1: Comparison of in-distribution segmentation performance in terms of DICE score.

| Model | $\lambda$ | Overall | Tumor Core | Edema | Enh. Tumor |
|---|---|---|---|---|---|
| MCDO U-Net | - | $0.576 \pm 0.15$ | $0.308 \pm 0.25$ | $0.557 \pm 0.23$ | $0.441 \pm 0.31$ |
| VI U-Net | - | $0.576 \pm 0.17$ | $0.313 \pm 0.27$ | $0.549 \pm 0.24$ | $0.442 \pm 0.34$ |
| Prob. U-Net | - | $0.537 \pm 0.21$ | $0.304 \pm 0.27$ | $0.468 \pm 0.33$ | $0.381 \pm 0.35$ |
| PhiSeg | - | $0.564 \pm 0.18$ | $0.326 \pm 0.28$ | $0.533 \pm 0.28$ | $0.401 \pm 0.32$ |
| SSNs | - | $0.576 \pm 0.14$ | $0.329 \pm 0.25$ | $0.561 \pm 0.23$ | $0.415 \pm 0.30$ |
| MH | 0.0 | $0.572 \pm 0.16$ | $0.316 \pm 0.26$ | $0.554 \pm 0.22$ | $0.419 \pm 0.32$ |
| MH | 0.5 | $0.590 \pm 0.14$ | $\mathbf{0.351 \pm 0.25}$ | $0.580 \pm 0.21$ | $0.430 \pm 0.31$ |
| MH | 1.0 | $0.582 \pm 0.15$ | $0.339 \pm 0.27$ | $0.567 \pm 0.21$ | $0.425 \pm 0.32$ |
| VI MH | 0.0 | $0.582 \pm 0.15$ | $0.328 \pm 0.25$ | $0.574 \pm 0.22$ | $0.427 \pm 0.31$ |
| VI MH | 0.5 | $0.584 \pm 0.14$ | $0.331 \pm 0.25$ | $\mathbf{0.588 \pm 0.21}$ | $0.418 \pm 0.32$ |
| VI MH | 1.0 | $\mathbf{0.598 \pm 0.15}$ | $0.339 \pm 0.25$ | $0.584 \pm 0.22$ | $\mathbf{0.469 \pm 0.31}$ |

We provide a thorough analysis of the proposed multi-headed Variational U-Net with respect to segmentation performance, uncertainty quantification and the OOD behavior. Further, we compare it with the state-of-the-art methods PhiSeg (Baumgartner et al., 2019), Prob. U-Net (Kohl et al., 2018) and SSNs (Monteiro et al., 2020).

**Experimental Setup:** We conduct all our experiments on the BRATS dataset with 200 training, 34 validation, and 101 test volumes. For all volumes, we sample the resolution

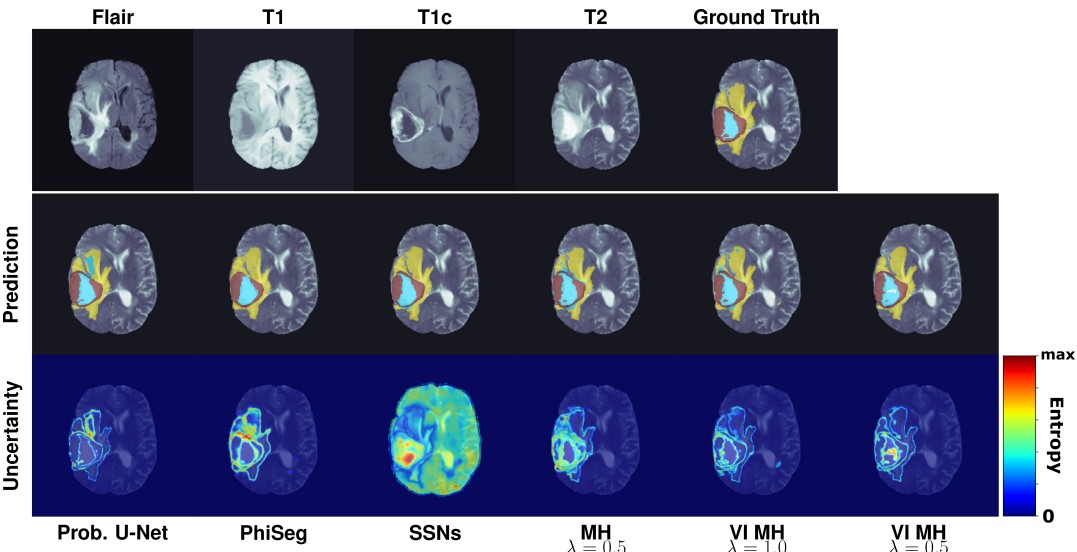

Figure 3: Qualitative comparison shows VI MH with $\lambda = 0.5$ delivers a interpretable and well-calibrated uncertainty estimate. The prediction consist of tumor core (blue), edema (yellow) and enhancing tumor (red).

to (256, 256, 155) voxels and spacing of (0.9375, 0.9375, 1.0). Each network is based on a slice-by-slice U-Net and trains with all four modes of MR acquisitions (FLAIR, T1, T1c, T2) as feature channels. To simulate OOD shift, we deploy four different transformations from the *TorchIO* library. We base the augmentations on artifacts that are observed regularly for MR images: Motion, Ghosting, Noise, and Spikes in the k-space. All experiments are run on an NVIDIA 1080Ti. Further details are provided in the supplementary material, and a PyTorch implementation is available on https://github.com/MECLabTUDA/VIMH.

**Quantitative Comparison and Ablation Study:** We compare the models in terms of segmentation performance for the three types of tumor tissues using subject-wise DICE score in Table 1. The Overall column describes the performance over all classes, including background. MH $\lambda = 0.5$ has the best tumor core segmentation and highest overall score for all MH models. However, the VI MH has the best enhancing tumor and overall segmentation performance with $\lambda = 1.0$. The edema is best segmented by the VI MH $\lambda = 0.5$. While Prob. U-Net has the worst performance of the models compared, PhiSeg can improve upon them with their multi-resolutional approach. The SSNs applied to the MCDO U-Net maintain the performance, but the training process is quite unstable.

**Qualitative Results:** The foremost criterion of uncertainty quantification is its clinical meaningfulness that establishes the clinician's trust in the segmentation model. For the evaluation of uncertainty, observation of the qualitative results is crucial. Figure 3 shows a sample slice along with the prediction and estimated uncertainty of the relevant models. While all models produce proper segmentation, the uncertainty estimation of Prob. U-Net only marks the edges as uncertain and even predicts falsely segmented tumor core without any uncertainty. PhiSeg can improve upon this quite significantly with its multi-resolution approach. However, the model provides an overly smooth segmentation. The uncertainty of PhiSeg does not provide any indication of these missing details e.g. in the lower part

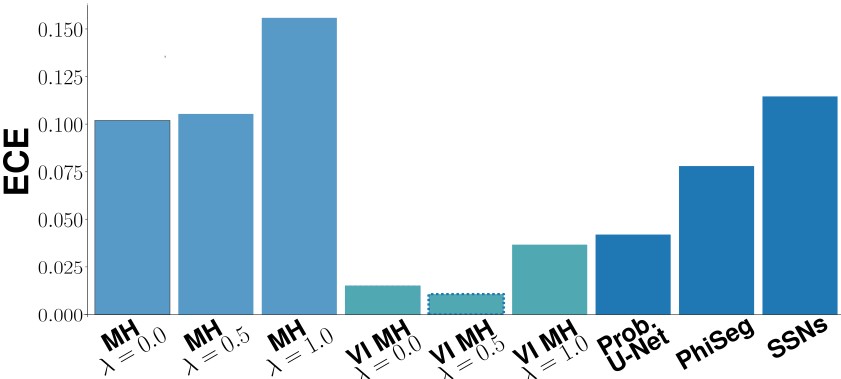

Figure 4: Expected Calibration Error: MHs lack in calibration due to limited amount of available samples. The VI helps the MH greatly in calibration.

of the tumor core. While MH $\lambda = 0.5$ produces great overall segmentation performance in Table 1, we can see the over-segmentation of the tumor core paired with a blurry uncertainty mask (Figure 3). Due to blurry uncertainty, an interpretation of model failure and false segmentation is impossible. The VI MH models (primarily $\lambda = 0.5$) provide potentially clinically relevant uncertainty quantification. The VI MH $\lambda = 0.5$ falsely does not segment part of the tumor core. However, the uncertainty mask is providing clear evidence that something is wrong at the exact location. The aleatoric uncertainty of the SSNs does not offer helpful information for a clinician as it describes the uncertainty inherent in the image data rather than estimating epistemic uncertainty. Incorporating the head-wise loss (VI MH $\lambda = 0.5$) produces good qualitative and quantitative results and Figure 4 shows a better calibration error compared to $\lambda = 1.0$.

**Calibration:** The segmentation should be analyzed in relation to the uncertainty quantification. The Expected calibration error (ECE, (See Appendix C) in Figure 4 shows that Prob. U-Net and VI MHs have good calibrations. PhiSeg and SSNs show worse calibration than Prob. U-Net. Further, the figure illustrates that the VI and integration of head-wise loss aid in calibration. The VI MH offers qualitatively much better uncertainty, and with the addition of the head-wise loss ($\lambda = 0.5$) the model improves further.

Table 2: Results under four different OOD shifts in DICE Score

| Model | Avg | Motion | Spike | Ghosting | Noise |
|---|---|---|---|---|---|
| **MCDO U-Net** | 0.500 | $0.534 \pm 0.16$ | $0.461 \pm 0.17$ | $0.493 \pm 0.15$ | $0.511 \pm 0.15$ |
| **VI U-Net** | 0.535 | $0.571 \pm 0.17$ | $0.510 \pm 0.16$ | $0.506 \pm 0.15$ | $0.554 \pm 0.15$ |
| **Prob. U-Net** | 0.389 | $0.273 \pm 0.05$ | $0.483 \pm 0.18$ | $0.273 \pm 0.05$ | $0.527 \pm 0.20$ |
| **PhiSeg** | 0.517 | $0.564 \pm 0.18$ | $0.446 \pm 0.18$ | $0.530 \pm 0.19$ | $0.526 \pm 0.18$ |
| **SSNs** | 0.509 | $0.574 \pm 0.14$ | $0.463 \pm 0.16$ | $0.487 \pm 0.12$ | $0.513 \pm 0.15$ |
| **VI MH $\lambda = 0.0$** | 0.540 | $0.577 \pm 0.14$ | $0.498 \pm 0.15$ | $0.533 \pm 0.14$ | $0.551 \pm 0.13$ |
| **VI MH $\lambda = 0.5$** | 0.548 | $0.578 \pm 0.14$ | $\mathbf{0.520 \pm 0.15}$ | $0.533 \pm 0.13$ | $\mathbf{0.559 \pm 0.13}$ |
| **VI MH $\lambda = 1.0$** | **0.552** | $\mathbf{0.593 \pm 0.14}$ | $0.515 \pm 0.15$ | $\mathbf{0.547 \pm 0.14}$ | $0.551 \pm 0.14$ |

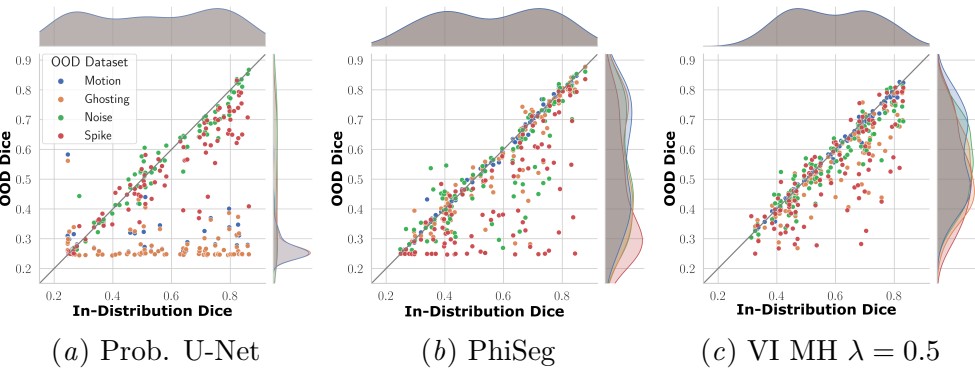

(a) Prob. U-Net      (b) PhiSeg      (c) VI MH $\lambda = 0.5$

Figure 5: DICE score for each patient on test set vs. OOD shifted test set. Spike artifacts affect the segmentation performance most. Motion artifacts are close to all models in-distribution performance, except Prob. U-Net.

**Robustness to OOD shift:** An important property for the deployment of segmentation and uncertainty quantification in practice is the robustness to OOD shifts. Figure 5 shows the performance of each model under OOD shifts. Prob. U-Net segmentation quality collapses when distribution shifts similar to artifacts from Motion and Ghosting occur. PhiSeg exhibits decent segmentation scores across all OOD shifts. However, Spike artifacts can cause failure, even for samples with high in-distribution performance. VI MH $\lambda = 0.5$ maintains good segmentations across all OOD shifts. Table 2 supplements those observations and shows that the VI MH $\lambda = 1.0$ is still the best performer in terms of average over the distribution shifts. However, VI MH $\lambda = 0.5$ can outperform on Spike and Noise artifacts. Further, VI MH $\lambda = 0.5/\lambda = 1.0$ are statistically similar on the OOD test data.

## 5. Conclusion

Good uncertainty calibration and out-of-distribution performance for medical segmentation are essential to ensure safe and robust applications. We introduce an efficient uncertainty-aware interpretation of U-Net. By combining VI and Deep Ensembles, we improve segmentation performance and uncertainty quantification. Our proposed multi-headed Variational U-Net incorporates the desirable properties of Deep Ensembles and VI, achieving a good segmentation performance, uncertainty calibration and robustness under distribution shift.

In our work, we selected an appropriate number of heads and split point based on a compromise between computational demands and performance improvement. Placing MH deeper into the architecture may improve performance but drastically increase computational demands. The proposed approximation is general enough to be applied to most state-of-the-art neural networks with relative ease. The ability to approximate the posterior for medical image segmentation in a computationally efficient manner opens up the possibility of several clinical applications. Specifically, when critical downstream tasks such as radiotherapy or ablation planning depend on these initial segmentations, properly calibrated models increase the trust of the clinicians and approval boards. Uncertainty-aware interpretations of popular deep learning approaches take us one step closer to trusted widespread adoption of deep learning in clinical settings.

## Acknowledgments

This work was supported by the Bundesministerium für Gesundheit (BMG) with grant [ZMVI1-2520DAT03A].

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

## Appendix A. Implementation Details

We train all networks with a batch size of 11 and Adam optimizer with default parameters and a learning rate of $3e^{-4}$. We use $H = 12$ ensemble heads for the MHs, and the VI MHs have $H = 4$ ensemble heads due to memory restrictions. The MCDO network has a 0.5 Drop-Out rate on the last two convolutional layers. The VI models use a prior of $P(w) = \mathcal{N}(0, 0.1)$ and a temperature of $T = 1e^{-8}$. We draw three samples for training and nine samples for inference of all sampling-based methods. We use our U-Net architecture (Figure 2 (a)) as a backbone for all reference implementations for a fair comparison.

**Prob. U-Net:** We train with all the parameters of Kohl et al. (2018), but drop all augmentation except the usage of pseudo-labels for each class with equal weighting. Further, we adapt the prior and posterior networks to fit our U-Net architecture.

**PhiSeg:** We reduce the model to four resolutions and two latent levels, adapting it to our U-Net architecture. In addition, we deploy the augmentations of flips, rotation, scaling, and pseudo-labels to stabilize the training, giving a clear advantage over all other models.

**SSNs:** We tested the Deep-Medic architecture (Monteiro et al., 2020) but find better results by using our pre-trained MCDO U-Net with deactivated Drop-Out as a base. We removed the last layer and replaced it with SSNs with ten low-rank factors.

**OOD Transformations:** We generate four different distribution shifts of the test dataset to evaluate the proposed approaches. Each dataset represents a specific artifact that can occur during the acquisition of MR images: Motion, Ghosting, Noise, and Spikes in the k-space. To simulate the transformation we use the implementation of the *TorchIO* library (Pérez-García et al., 2021). For the Motion shift, we use a maximum rotation of 20 and a translation of 12. Ghosting is deployed in three dimensions with one to ten ghosts. On the input range $[0, 1]$, we deploy Gaussian Noise with a mean in the range of $[0.2, 0.5]$ with a standard deviation ranging between $[0.2, 0.5]$. Lastly, we use Spike artifacts in the k-space with one to three spikes and intensity from one to three times the maximum in the frequency spectrum.

## Appendix B. Functional differences

While Fort et al. (2019) show the functional differences between full-size ensemble members for classification, it is unclear whether parameter-sharing approaches can represent a divers set of functions. Therefore, we indicate the existence of different local optima for the heads on a segmentation. We analyze the functional differences between ensemble heads in two ways. First, we compare their segmentation differences for different $\lambda$'s. Since most voxels, e.g. air, within the BRATS dataset have only one optimum, we filter the segmentations for regions that have a multi-modal loss landscape. We introduce a threshold $t$ to exclude higher confidence quantiles. This identifies examples, where the uncertainty estimation is most helpful to the user. On the extracted regions we calculate the dissimilarity $d$ between all ensemble heads and the ensemble prediction (E) as follows: $d = 1 - DICE$. This helps to identify if ensemble heads tend to represent the same function on the loss landscape.

Figure 6 shows that training with $\lambda = 0.5$ increases the dissimilarity between the ensemble heads compared to $\lambda = 0$. The $\lambda = 1.0$ MH shows extreme dissimilarities due to bad calibration (Figure 4). In Figure 7 (a) for the VI MH $\lambda = 1.0$, the similarity increases

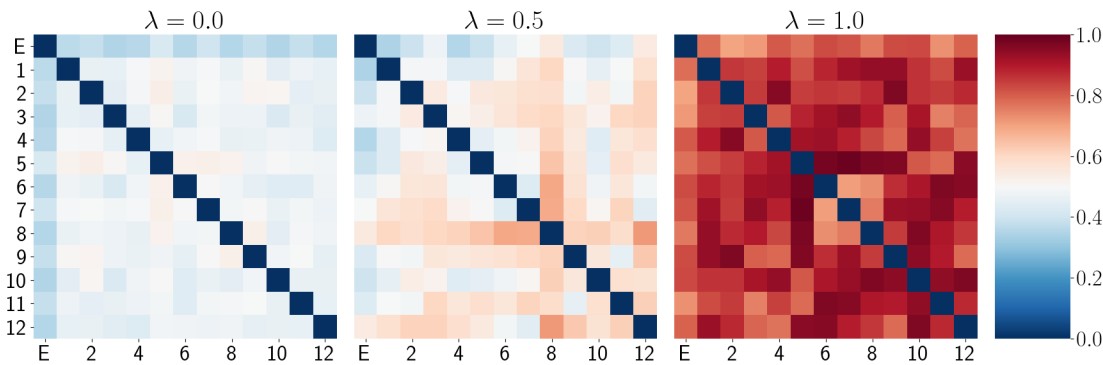

Figure 6: The DICE dissimilarity between the 12 heads and ensemble solution (E) with $t = 0.99$ to exclude high confidence predictions. The first row and column is dedicated to the ensemble solution marked by E. From left to right the MH trained with $\lambda = \{0, 0.5, 1\}$

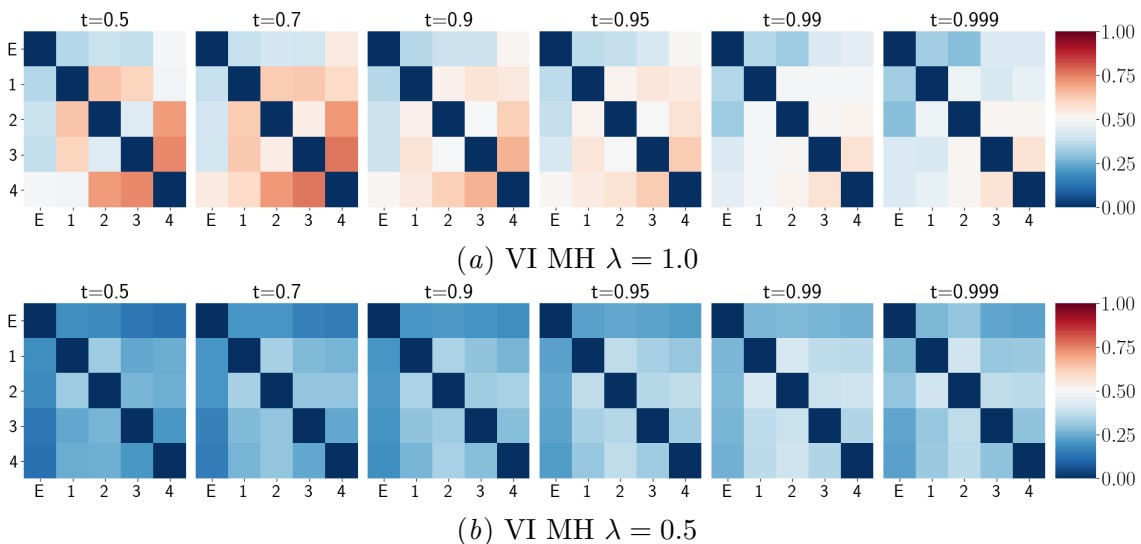

Figure 7: A combination of both loss terms is beneficial. Increasing uncertainty threshold $t$ (from left to right) shows a better stability in confusion matrices for $\lambda = 0.5$ (b) than $\lambda = 1$ (a).

for high uncertainties. This is not a desired behavior, as we want the uncertainty to correlate well with the performance of the model. For the VI MH $\lambda = 0.5$ in Figure 7 (b), the ensemble heads converge to more dissimilar models, indicating local optima for higher uncertainties. Together with the good DICE score this makes for a desirable model.

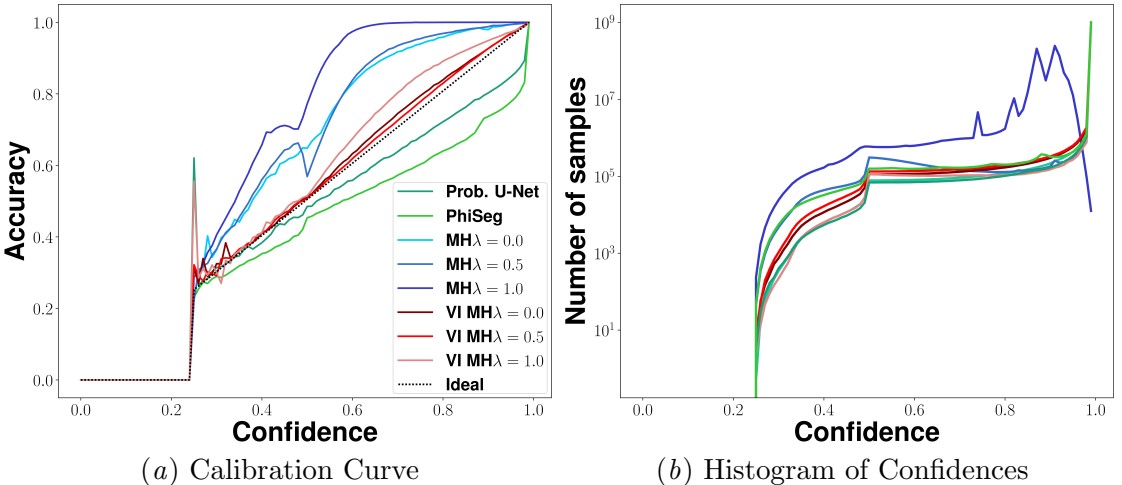

($a$) Calibration Curve  ($b$) Histogram of Confidences

Figure 8: (a) Calibration Curve: The VI MH Ensemble ($\lambda = 0.5$) a has the best Expected Calibration Error. (b) Histogram of predicted confidences on a log scale.

## Appendix C. ECE and Calibration curve

The ECE is defined as:

$$ECE = \sum_{b=1}^{B} \frac{n_b}{N} |acc(b) - conf(b)|$$

with $N$ the total number of voxels and $n_b$ the number of voxels per bin. The bins $B = 100$ are equally spaced between 0.25 (1/number of classes) and 1.0.

In a real-world application, we want to know what our model thinks how accurate its prediction is. For example, if the confidence in a prediction is 60%, we want this prediction to be accurate with a 60% chance. The ECE tries to measure this relationship. One crucial detail is the re-weighting of the bins to give every sample the same contribution. The re-weighting is especially important for low confidence predictions with only a few samples. Such a bin would significantly influence the expected calibration error more than intended (Nixon et al., 2019). A graphical interpretation of the ECE score can not represent this problem. In Figure 8, we compare the calibration curves of our models with Prob. U-Net and PhiSeg. While all multi-headed models are not calibrated, the VI can significantly improve their performance. Prob. U-net and PhiSeg represent an overconfident curve by being less accurate than the prediction confidence implies.

## Appendix D. More Qualitative Results

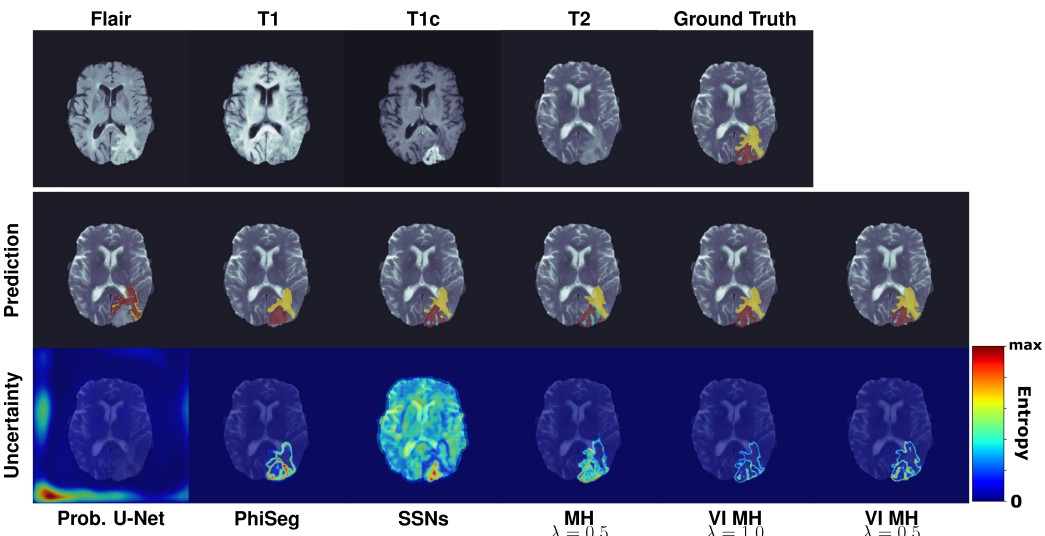

Figure 9: Qualitative comparison shows Prob-U-net to fail to deliver a prediction with an hard to interpret uncertainty estimation.

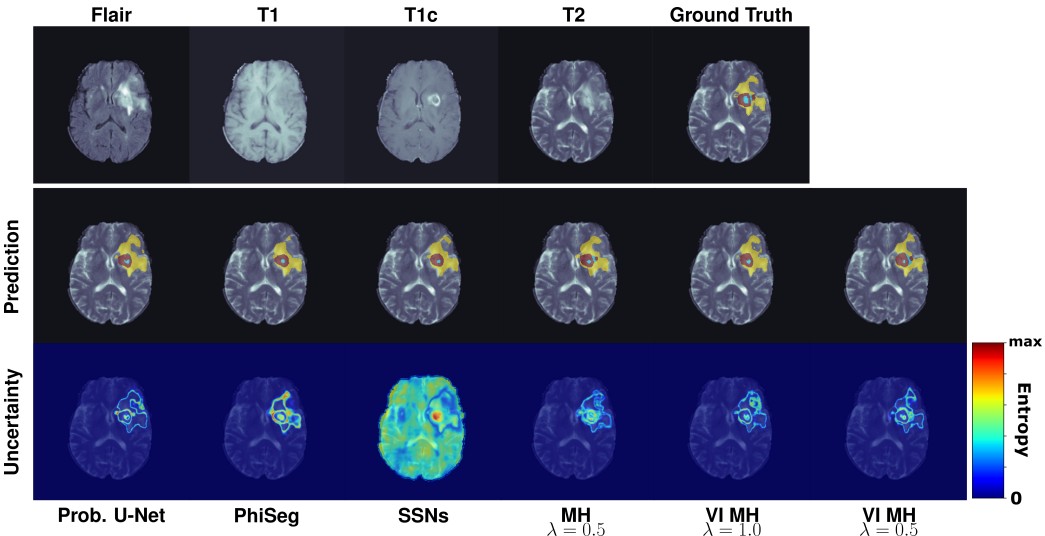

Figure 10: Qualitative comparison shows MH with $\lambda = 0.5$ deliver a hard to interpret uncertainty estimation.

## Appendix E. Significance Test

We test our methods segmentation performance significance on two hypotheses:

$(H_0)$ VI MH $\lambda = 0.5$ is not significantly worse than Prob. U-Net, PhiSeg and SSNs.

$(H_1)$ VI MH $\lambda = 0.5$ is significantly better than Prob. U-Net, PhiSeg and SSNs.

We perform the tests on both in-distribution (ID) data as well as on the OOD data. We consider all four distribution shifts as a combined dataset for the OOD data. As a result, we see that we are significantly better than Prob. U-Net and PhiSeg on ID data. We can not find our method worse or better than SSNs. On OOD data, we are significantly better compared to the competitors (See Table 3).

Table 3: Significance test results

| VI MH $\lambda = 0.5$ | Prob. U-Net | | SSNs | | PhiSeg | |
|---|---|---|---|---|---|---|
| | $H_0$ | $H_1$ | $H_0$ | $H_1$ | $H_0$ | $H_1$ |
| **ID dataset** | 0.995 | 0.003 | 0.839 | 0.08 | 0.912 | 0.044 |
| **OOD datasets** | 1.0 | 0.001 | 1 | 0.001 | 1 | 0.001 |

