# OpenReview forum: "Practical uncertainty quantification for brain tumor segmentation"
_MIDL.io/2022/Conference — MIDL 2022_

### Official Review · Reviewer_zBLu · 2022-01-11

**Confidence:** 5
**Preliminary Rating:** 4
**Recommendation:** Oral, Poster

**Summary:**

This paper describes a strategy based on a multi-headed variational u-net design for image segmentation that is aimed at dealing with issues of global instance exploration  (i.e. out-of-distribution shifts) while handling uncertainty during a segmentation task. The final design is applied to the BRATS2018 brain tumor public dataset with experimentation on different data distribution shifts with promising results, in comparison to some other SOTA methods.

**Strengths:**

The integration of  a u-net based architecture, global exploration and a variational inference design on the heads into the basic multi-headed design of the EnsembleNet is new and interesting and offers potentially improved performance on a variety of medical image segmentation tasks.

The detailed design of the VI (variational inference) modules is nicely done, including the introduction of the local re-parameterization “trick” originally proposed by Klingma et al.   In addition, the design of the two losses (one over all heads and one for each head), using Bayesian convolutions throughout the underlying U-Net for the overall VI MH (multihead)  architecture is an apparently unique and useful approach.  The ideas are reasonably motivated and the reduced number of parameters and relatively sped-up training is appreciated in comparison to the traditional ensemble approaches.

The test results shown in Table 1 in terms of DICE scores show decent performance improvement over reasonable alternatives (VI U-Net, Probabilistic U-Net, etc) using an appropriate training/ validation/ testing split on the BRATS dataset.   The out of distribution (OOD) testing and performance comparisons in Table 2 and figure 5 is very helpful and more clearly shows the advantages of the proposed method under these different conditions (motion, spikes, ghosting, noise)


**Weaknesses:**

It is rather difficult to understand how the proposed architecture affects the final uncertainty in the segmentations either from the standard deviations of the DICE test results in Table 1 or in the qualitative results shown in figure 3 (i.e. the Prob U-Net uncertainty result doesn’t look much different that the VI MH  results). It would be helpful for the authors to more clearly point out the differences (and/ or point to the differences on the images in figure 3). They try to do this in section 4 in describing the different errors, but (as noted by the authors) it is all hard to interpret ---although quantitative calibration error is more clearly separated.

Also, there is a concern that the design is tuned quite specifically for use with the BRATS2018 dataset only. In other words, will it easily extend for application to other datasets/ segmentation problems?

**Deanonymize Review:**

no

**Detailed Comments:**

More discussion on the meaning of improvements in uncertainty quantification due to the new design would be helpful. Perhaps more details on how the expected calibration error is obtained and what it means related to figure 3 or Table 1 would be helpful.

**Paper Type:**

both

**Questions To Address In The Rebuttal:**

(first part similar to last section) More discussion on the meaning of improvements in uncertainty quantification due to the new design would be helpful. Perhaps more details on how the expected calibration error is obtained and what it means related to figure 3 or Table 1 would be helpful and/or arrows pointing to the meaningful differences in uncertainly in figure 3 between the Prob U-net and PhiSeg vs the VI MH design.

Some discussion as to whether the design easily extends beyond only using the BRATs data would be helpful.


**Special Issue:**

yes

---

### Official Review · Reviewer_JkP8 · 2022-01-24

**Confidence:** 4
**Preliminary Rating:** 4
**Recommendation:** Oral

**Summary:**

The manuscript proposes a variational inference multi-head network to improve on uncertainty prediction and calibration applied to multi-class tumour segmentation using the BRATS challenge in a context where only one rater has performed the annotations
The experiments are composed of an ablation study (addition of the VI part for multi head) and comparison to state of the art solutions (Prob-Unet, Monte-Carlo DropOut, Stochastic segmentation network and PhiSeg. Performance is also evaluated in the context of image artefacts (considered as out of distribution examples)

**Strengths:**

- Well written paper with a clear motivation and substantial experimental design
- Good assessment of existing solutions and fair benchmarking
- Investigation of multiple aspects of the performance both in terms of segmentation performance and uncertainty modelling



**Weaknesses:**

- The loss function for the MH only model is unclear (how can the $\lambda$ value play a role) and where the ground truth play a role
- The expected calibration error does not appear to be explained
- Some of the figures are difficult to read/interpret due to the choice of markers/colors

**Deanonymize Review:**

no

**Detailed Comments:**

The paper is very interesting and overall well written. It is unfortunate that some key details are hidden in the supplementary material in particular the discussion regarding the choice for $\lambda$.

Regarding the experiments, 4 VIMH is compared to 12 MH without VI. For fairness, the same number of heads should probably have been chosen and an additional experiment would have been to look into the effect of the number of heads on the performance. The authors state that VI doubles the number of parameters so one would have expected the same ratio across number of heads.

Regarding Figure 2 and Figure 5, some colours are extremely similar making their distinction difficult. For Figure 5 a suggestion would be to choose also different markers

In terms of the loss function, it is not clear where the ground truth segmentation is used for the prediction.

As a remark, considering the proposed artefacts as fully OOD may be a bit exaggerated as some degree of artefacts are probably observed across the chosen sample and this could explain the relatively high performance observed across models. Further, the OOD nature of the artefacts applied would probably be related to their strength.

It is difficult from Table 2 to assess whether any of these results are statistically significant.

**Paper Type:**

methodological development

**Questions To Address In The Rebuttal:**

It would be good to address the comments related to the loss function and the explanation of the ECE.
Could you also address the question of the statistical differences in Table 2 - a figure here may be useful

**Special Issue:**

no

---

### Official Review · Reviewer_h6at · 2022-01-25

**Confidence:** 4
**Preliminary Rating:** 3

**Summary:**

1. This paper explores adding variational inference to a multi-headed U-Net to improve model calibration and uncertainty estimation of a U-Net segmentation.
2. Experiments on BRATS-18 dataset shows the proposed model can improve segmentation performance and uncertainty quantification over five baseline methods.

**Strengths:**

1. The experimental designs are reasonable and complete: segmentation performance (dice score), calibration performance (ECE), and synthetic OOD performance (dice score).
2. The insight of this paper is interesting.

**Weaknesses:**

1. Though I think this paper provides some insights, I am worried about the novelty of the paper. The current presentation of the paper is incremental to a U-Net segmentation. The variational inference is not new and the multi-headed approach is also not new. I would encourage the author to not only do experiments to show that adding MH VI can improve U-Net segmentation, but also for other SOTA backbones. There is no evidence indicated in this paper that the idea can apply to other SOTA segmentation models.
2. Some experimental details are missing. Please see comments in the next sections.
3. The motivation of this paper is not well-established.

**Deanonymize Review:**

no

**Detailed Comments:**

1. The author might consider adding significance test in table 1 for better interpretation. Some of the results are very similar and a significance can tell a better story.
2. To improve the strength of the paper, it is better to include other SOTA architecture backbones other than U-Net to test the benefit of VI and MH.
3. It is difficult to get any useful information from “We conduct all our experiments on the BRATS dataset with a 60% training, 10% validation and 30% test split.” The author may want to use absolute numbers for reporting this data split.
4. It might be better to include more information for this sentence “we sample the resolution to 256 × 256 × 155 voxels”. For example, the original image is of size (xxx), voxel spacing (yyy), downsampling to (zzz) with what methods (linear mapping, crop outliers, intensity range etc.)
5. There is no mention about the appendix B in the main manuscript. How does it relate to the main story? The author might consider adding a few sentences to explain it.
6. I do not know how to interpret Figure 3. I do not understand why “VI MH with λ = 0.5 delivers an interpretable and well-calibrated uncertainty estimate”. For me, what I see in VI MH $\lambda$=0.5 has an obvious wrong prediction area inside the blue highlighted region. I do not think the descriptions in qualitative results paragraph convince me.
7. The author can draw a figure details about the dice score and ECE with respect to $\lambda$ ranging from 0 to 1 step with 0.1 in Appendix. In this way, it tells a better story than the current paper that only includes ($\lambda$=0, 0.5, 1).
8. Honestly, I did not find Figure 1 informative. It is the conceptual ideas without experimental results back-up. I would suggest the authors to use an example results to validate the points of variational methods capture the local minimal and multi-headed ensemble identifying different modes. This could be done by taking a close look at a local patch inside the image and visually and quantitatively demonstrate the points.

**Final Rating After The Rebuttal:**

2: Weak Reject

**Justification Of The Final Rating:**

Based on the answers and results the authors presented in the paper and the rebuttal, the paper is not ready to be accepted.

The good point is the author's OOD experiment design.

For weaknesses that can be strengthened are:

1. The most weak point of this paper is the performance. The author reported a performance at round 55%\~60% dice score in the paper, however most of the methods in BRATS-18 dataset leaderboard (https://www.cbica.upenn.edu/BraTS18/lboardValidation.html) and the workshop paper summary (https://arxiv.org/abs/1811.02629) are above 80% in terms of dice score. The author's main arguments are because the data split, data augmentation, and limited computing resources. Note that the official BRATS-18 dataset contains 285 images for training, 66 for validation, 191 for testing. The author reported their data split as 200 images for training, 57 for validation and 101 for testing. On the other hand, when the authors tried the method **NVDLMED** (https://arxiv.org/pdf/1810.11654.pdf#:~:text=This%20year%2C%20BraTS%202018%20training,input%20image%20size%20is%20240x240x155) that won brats-18 first place in 2018 in their "brats-18 dataset split training setting",  they reported a performance at 31.5%, which is very worse. Note that the **NVDLMED** paper reported a performance of 85.94% in 66 image validation dataset and of 82.19% in final 191 image testing dataset. These validation and testing dataset do not have public available labels at the MICCAI challenge period. **NVDLMED** has been cited 473 times and have lots of open-sourced implementations to verify the performance. It is performance could not be as terrible as 31.5% in BRATS-18 dataset. Hence, it raises a question that whether the authors have trained reasonable models in BRATS-18 dataset. Thus, it is not possible to conclude whether the proposed MH VI is useful or not.

2. The current novelty of the paper is incremental to a 2D U-Net and rather weak. It just says that MH + VI can be used in a 2D U-Net to make it more robust to OOD. To the reviewer's point, by adding more experiments on other well-known backbone structures, it can tell the story whether MH + VI is a correct design or not and whether the performance gain is reasonable or not.

3. Based on the reliability diagram the author reported. The reviewer is confident that some of the models are not well trained and have performance issues.

4. There are missing details about the proposed approach and the fine-tuning of the other baseline methods, as well as other experimental details. The authors have added some of the details based on all three reviewers's comments. But that is far away from being a well-written paper.



**Paper Type:**

both

**Questions To Address In The Rebuttal:**

1. What is the definition of ECE in this paper? If the definition comes from probability calibration, there are bins that need to be specified. If there are other definitions, it should be defined either in the main context or the appendix. Also, ECE usually comes with reliability diagrams to show the probabilities distribution and whether the model is overwriting to probabilities or underfitting to probabilities.
2. What is the definition of uncertainty in this paper? Is it entropy or probability or variance? What is the quantitative value for $\sigma_{max}$ in Figure 3?
3. Can the VI+MH modules show benefits on other backbone architectures?
4. What does this sentence mean “Each network is based on a slice-by-slice U-Net”? Do you use a 2D U-Net instead of a 3D U-Net? The data is in 3D format so it makes sense to use 3D U-Net. The citation only contains 2D U-Net[1] not 3D U-Net[2].
5. Please provide detailed pre-processing steps about the data?
6. I briefly looked at the leaderboard of BRATS-18 (https://www.cbica.upenn.edu/BraTS18/lboardValidation.html) and the workshop paper (https://arxiv.org/abs/1811.02629). I did not find that the author’s reported performance close to the SOTA results. Can the authors comment on this?
7. I do not understand this sentence “We draw three samples for training and nine samples for inference of all sampling-based methods.” in Appendix A.


[1] Ronneberger, O., Fischer, P., & Brox, T. (2015, October). U-net: Convolutional networks for biomedical image segmentation. In International Conference on Medical image computing and computer-assisted intervention (pp. 234-241). Springer, Cham.

[2] Çiçek, Ö., Abdulkadir, A., Lienkamp, S. S., Brox, T., & Ronneberger, O. (2016, October). 3D U-Net: learning dense volumetric segmentation from sparse annotation. In International conference on medical image computing and computer-assisted intervention (pp. 424-432). Springer, Cham.

**Special Issue:**

no

---

### Meta-Review · Area_Chair_Vta5 · 2022-02-20

**Recommendation:** Accept (Poster)
**Confidence:** 4

**Metareview:**

The authors investigate the value of adding variational inference to a multi-headed U-Net in terms of the quality of predictive uncertainty estimation and out-of-distribution performance on the BRATS2018 segmentation challenge dataset.

The reviewers agree that the evaluation strategies---segmentation performance (dice score), calibration performance (ECE), and synthetic OOD performance (dice score)---are quite thorough, and empirical improvements over the competing methods are significant.

However, I would like to note that as Reviewer 1 has commented, the technical novelty might be limited as the contribution seems to be the addition of variational inference to a multi-headed U-Net architecture.

Overall, given strong empirical results, I would like to recommend acceptance. I would like to, however, encourage the authors to describe this technical contribution more precisely in the introduction by mentioning the appropriate prior works and describing how to combine them (e.g. multi-head U-Net + VI).

---

### Decision · Program_Chairs · 2022-02-28

Accept